# Effect of Mechanical Pruning on Olive Yield in a High-Density Olive Orchard: An Account of 14 Years

**António Dias [1,\*], José Falcão [2], Anacleto Pinheiro [1]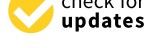 and José Peça [1]**

1   MED—Mediterranean Institute for Agriculture, Environment and Development & Departmento de Engenharia Rural, Escola de Ciências e Tecnologia, Universidade de Évora, Pólo da Mitra, Ap. 94, 7006-554 Evora, Portugal; pinheiro@uevora.pt (A.P.); jmop@uevora.pt (J.P.)
2   Torre das Figueiras Sociedade Agrícola Lda Apartado 23, 7450-909 Monforte, Portugal; cffjosefalcao@gmail.com
\*   Correspondence: adias@uevora.pt

**Abstract:** In Portugal, the study of mechanical pruning in olive orchards began in 1997. Trials were carried out in traditional orchards, and the obtained results revealed that mechanical pruning can contribute to the reduction in pruning costs without a reduction in yield. In 2005, the authors started an evaluation of mechanical pruning in high-density olive orchards. The trial was organised in a randomised, complete block design, with three replications. Four treatments were compared: T1—manual pruning, using chain saws in 2005, 2010, 2014, and 2017; T2—mechanical pruning, topping, and hedging (two sides) in 2014 and 2017, followed by manual pruning complement; T3—mechanical pruning, topping in 2005, topping and hedging (west side) in 2008 and 2012, topping and hedging (east side) in 2010, 2014 and 2017, summer topping in July 2015, and hedging (west side) in winter 2016; T4—mechanical pruning, topping in 2005, topping and hedging (two sides) in 2010, 2014, and 2017, and summer topping in July 2015. The average yield per tree for each treatment was evaluated. In the first period (5 years), no significant differences were found between treatments. In the second period (4 years), a greater frequency of mechanical pruning (T3) showed a lower yield, with significant differences with treatment T2. In the third period of 5 years, a greater pruning severity reduced the olive yield (2014). Manual pruning complement after mechanical pruning (T2) did not increase olive yield; however, the average yield over this period of time was similar. The olive yield was maintained over a period of 14 years by only applying mechanical pruning, without manual pruning complement inside the canopy.

**Keywords:** mechanical pruning; canopy topping; canopy hedging



## 1. Introduction

In Portugal, the scarcity of labour led to the use of a high level of mechanisation to prune olive groves. Tests carried out on traditional olive orchards have shown that the use of disc-pruning machines allows for high pruning rates, which can contribute to reducing pruning costs [1]. Refs. [2,3] also reported an important increase in the rate of pruning using disc-saw pruning machines. More recently, in olive trees trained as free-standing canopies with central leaders, the use of mechanical cuts in the side faces of the canopy allows increased pruning rates in comparison with manual pruning [4]. A high-density olive orchard trained to vase [5] was also reported to have an increased pruning rate with the use of mechanical pruning. The use of a horizontal cut at the uppermost part of the canopy (topping) with lateral cuts in the sides of the canopy leads to a decrease in mechanical pruning rate [5] due to a higher number of the machine passes per tree row. The use of mechanical pruning + manual pruning complement leads to a decrease in the pruning rate [5]. Trials carried out by [3–6] are short-term tests, and the evaluation of pruning solutions requires a longer period of time to allow the definition of a long-term pruning strategy.

In Spain, a trial conducted in a traditional rain-fed olive grove from 1981 to 1997 revealed similar olive yield per tree in mechanised and manual pruned trees [7]. These authors also stated the importance of occasional manual selective pruning to clear excessive wood from trees that were subjected to mechanised pruning [8].

Results obtained by [8] revealed that, after mechanical pruning (topping at the uppermost part of the canopy), trees can be kept for 8 years without any significant loss in olive yield or effect on harvesting efficiency. For a period of nine campaigns, selective manual complement, simultaneously used with mechanical pruning, brings no practical advantage in terms of yield [9].

In a second period, selective manual pruning that complements mechanical pruning, particularly in the year following mechanical pruning, should be regarded as a potentially important technique since it may contribute to higher yields [9].

In a denser (8 m × 4 m) rain-fed olive grove, assessments made by [7] over 8 years revealed that mechanical pruning and manual pruning resulted in similar olive yields.

In an irrigated, high-density olive orchard, a trial made over 8 years with selective manual pruning, mechanical pruning, and mechanical pruning complemented by a manual clearing of excessive wood, revealed that the mechanised pruning treatments, with or without manual complement, resulted in similar olive yields, which were significantly higher than the yield of the selective manual pruning [7] (cited in [8]).

The increase in the area of irrigated high-density (HD) olive orchards led the authors to evaluate the use of mechanical pruning in this kind of groves, which had been started in 2005 in an olive orchard of Picual variety.

This paper shows the results from the use of mechanical pruning on olive yield over 14 years, which can contribute to the definition of a mechanised pruning strategy for HD olive orchards.

## 2. Materials and Methods

### 2.1. Olive Orchard and Cultural Practices

The HD olive orchard used in the trial was established in 1996 in Herdade da Torre das Figueiras in the Alentejo region of southern Portugal (lat. 39°03′34.04″ N; 07°28′22.00″ W). This drip-irrigated, HD olive orchard of the Picual variety was installed in an array of 7 m × 3.5 m.

The orchard was planted on Chromic Luvisol soil (FAO). This region is semi-arid, with a strong continental influence and an annual mean rainfall of 500 mm concentrated in the winter.

The orchard is drip-irrigated twice a week, from May until October, receiving annually an estimated volume of 1500–2000 m$^3$ per hectare.

The HD olive orchard was sprayed to control olive leaf spot (Flusicladium *oleaginum (Castagne) Ritschel & U.Braun*), olive moth (Prays oleae Bernard), olive fly (*Bactrocera oleae Gmelin.*), and olive anthracnose (*Colletotrichum acutatum Simmons* or *Colletotrichum gloeosporioides Penz.*). Glyphosate was used for weed control in the rows, although a shredder machine was also used between rows. A yearly average of approximately 80 units of nitrogen, 30 units of phosphorus, and 50 units of potassium were applied to the soil by drip irrigation.

### 2.2. Equipment

Mechanical pruning was performed using a Reynolds & Oliveira Ltd. (R&O) disc-saw pruning machine mounted (Figure 1) on a front loader of a 97 kW (DIN) 4WD agricultural tractor [1,7].

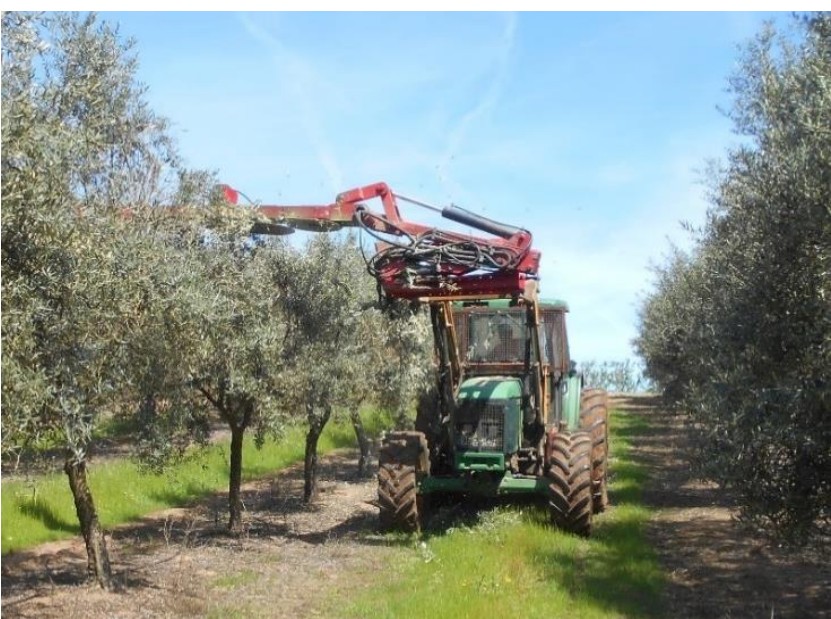

**Figure 1.** Mechanical pruning (topping).

The manual pruning complement to the mechanical pruning was executed with the use of telescopic chain saws.

Harvesting was performed using an 88 kW (DIN) self-propelled multidirectional Orchard Machinery Corporation (OMC) trunk shaker [2], except in the period between 2014 and 2017, when a side-row continuous canopy shaking harvester (SRCCSH) was employed [10].

### 2.3. Treatments

In a randomised, complete block design with three replications, 4 treatments (T1, T2, T3, and T4) were compared, leading to 12 one-line plots, with 30 trees per plot. The treatments were as follows: T1—manual pruning, using chain saws in 2005, 2010, 2014, and 2017; T2—mechanical pruning, topping and hedging the two sides of the canopy in 2014 and 2017, followed by a manual pruning complement to remove wood suckers inside the canopy; T3—mechanical pruning, topping the canopy parallel to the ground in 2005, topping the canopy parallel to the ground, and hedging the west side of the canopy in 2008 and 2012, topping the canopy parallel to the ground and hedging the east side of the canopy in 2010, 2014, and 2017, summer topping the canopy in July 2015 (summer pruning), and hedging the west side in winter 2016; T4—mechanical pruning, topping the canopy parallel to the ground in 2005, topping and hedging the two sides of the canopy in 2010, 2014, and 2017, and topping the canopy in July 2015 (summer pruning).

Common to all treatments were manual pruning to eliminate hanging branches in 2006 and mechanical pruning and topping the canopy parallel to the ground to eliminate 0.50 m, in 2007.

The sequence of pruning interventions made in this trial is presented in Figure 2.

| Trat | 2005 | 2006 | 2007 | 2008 | 2009 | 2010 | 2011 | 2012 | 2013 | 2014 | 2015 | 2016 | 2017 | 2018 |
|---|---|---|---|---|---|---|---|---|---|---|---|---|---|---|
| T1 | Manual | Hanging branches | [topping] | | | Manual | | | | Manual | | | Manual | |
| T2 | No pruning | Hanging branches | [topping] | | | | | | | [both sides] + Manual compl. | | | [both sides] + Manual compl. | |
| T3 | [topping] | Hanging branches | [topping] | [topping] | | [topping] | | [topping] | | [topping] | [topping] Summer | [topping] | [topping] | |
| T4 | [topping] | Hanging branches | [topping] | | | [both sides] | | | | [both sides] | [topping] Summer | | [both sides] | |

Legend: manual—manual pruning; manual compl.—manual pruning complement; summer—summer pruning; hanging branches—removed hanging branches; [icon] —mechanical pruning (topping); [icon] —mechanical pruning (topping + hedging west canopy side); [icon] —mechanical pruning (topping + hedging east canopy side); [icon] —mechanical pruning (topping + hedging both canopy sides).

**Figure 2.** Sequence of pruning interventions.

*2.4. Assessments*

The time spend by pruners in each plot was registered. In mechanical pruning, it was quantified the time needed to cut each side of the canopy and to make the topping (including turning time), in each plot.

From 2005 until 2013, and also in 2018, olive harvesting was made by trunk shaker complemented by manual harvest with poles. Olives were collected in nets manually placed in the ground above each tree canopy. Olive yields were measured by weighing the olives harvested plot by plot.

In the period between 2014 and 2017, harvesting was performed by using the SRCCSH prototype. The mass of olives caught with SRCCSH was measured by weighing the bags from each plot. The evaluation of the mass of olive removed but not caught by the harvester was carried out by weighing the fruits collected on canvas, placed under a group of 3 olive trees at 3 locations randomly selected in each plot. To quantify the mass of olives not removed by the harvester, all trees in each plot were vibrated by a trunk shaker complemented by manual harvest with poles.

Total yield per tree was obtained by adding the mass of olives caught by the harvester (in the bags) to the mass of olives dropped to the ground (on the canvas) plus the mass left on the tree.

During this trial, fruit drop before the harvest was small and, therefore, discarded.

One-way analysis of variance was performed on annual data, and GLM univariate analysis was used for average data, using IBM SPSS version 24 software. Mean separation was performed by Tukey HSD test at 5% and 10% significance levels.

## 3. Results

*3.1. Pruning Work Rates*

Figure 3 shows the manual pruning work rates registered during this evaluation. The higher work rate recorded in 2005 was associated with the smaller size of the olive trees

(orchard with 9 years). In 2010, the larger size of the trees combined with a different type of pruning (elimination of branches developed lengthwise) led to a much lower manual pruning rate than in 2005. In 2014, there was an increase in the manual pruning rate, as the intervention was similar to that normally performed by the pruners (elimination of branches inside the canopy).

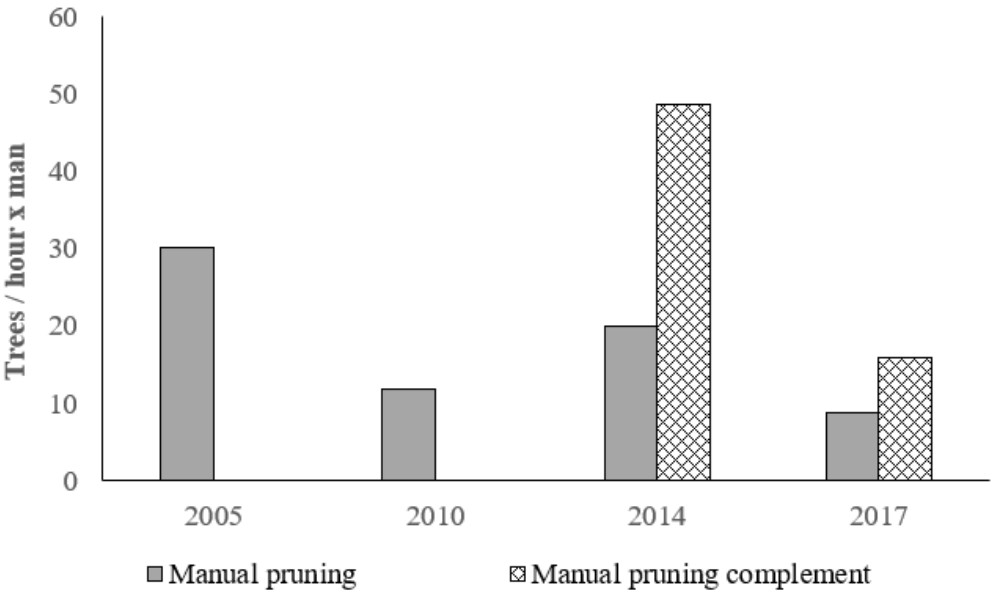

**Figure 3.** Average work rates of manual pruning.

The removal of these branches resulted in the exposure of the central part of the canopy to sunlight. This led to the emission of many suckers in this area of the canopy, which tended to fill it completely. In the next pruning intervention, if the objective was the removal of the suckers from the inner part of the canopy, a great number of cuts would be necessary, resulting in a reduction in the work rate, as shown by the results obtained in 2017.

To avoid low work rates of manual pruning complement, as reported in previous research studies [1,10], a clear definition of the features included in manual pruning complement, as well as continuously monitoring pruning workers, led to higher work rates than those obtained by strictly manual pruning intervention.

Figure 4 shows the working rate of the disc-saw pruning machine in each treatment. In 2005, it was necessary to carry out only one passage of the pruning machine per tree line, allowing a high work rate.

Topping and hedging on both faces (T4) corresponded to a working capacity of 155 to 175 trees per hour (2010–2017). Topping and hedging of only one side (T3) of the canopy (2010–2017) showed a slightly higher work rate (186 to 223 trees per hour) due to the non-productive return path required to restart the work in the same direction. In 2008, the work rate was clearly higher since the trees had a smaller dimension (only one pass by tree row), with thinner branches, which allowed the topping to be performed with a higher speed (2.9 kmh$^{-1}$, on average).

Summer topping (2015), despite being carried out at a lower working speed (1.0 kmh$^{-1}$, on average) than the winter topping (2.0 kmh$^{-1}$, on average), registered a greater working capacity, since only one machine pass was necessary for each row of trees, whereas two passes were required when performing winter topping due to wider canopies and limited cutter bar width.

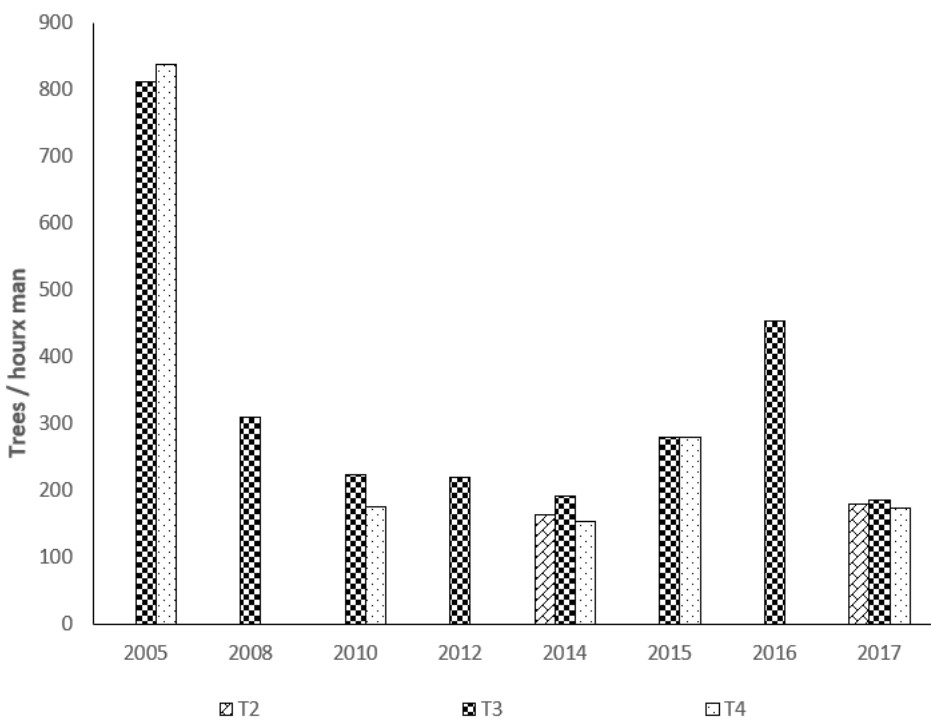

**Figure 4.** Average work rates (trees/hour × man) by treatment with the disk-saw pruning machine: T2—treatment 2; T2—treatment 3; T4—treatment 4.

*3.2. Pruning Costs*

Based on the working capacities presented in Figures 3 and 4, the pruning costs were determined. The following assumptions were considered: 70 EUR/h for contractor work charges for mechanical pruning; 70 EUR /man-day for contractor work charges for manual pruning; 7.5 h of effective work per day.

Figure 5 shows the cost of pruning in each year and by treatment, as well as the total value per treatment over the period of the trial (2005 to 2018).

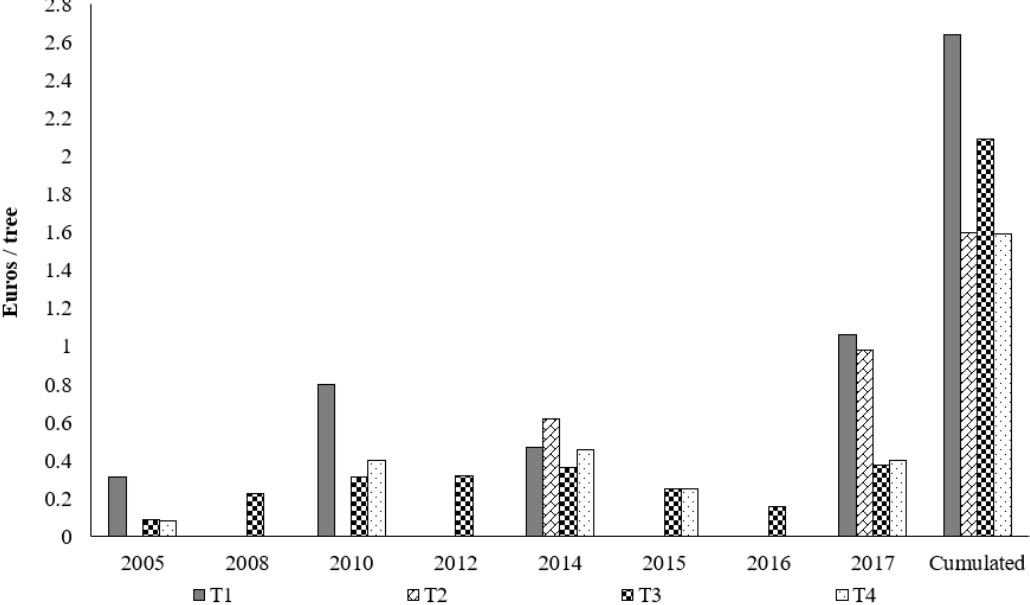

**Figure 5.** Pruning costs per treatment: T1—treatment 1; T2—treatment 2; T3—treatment 3; T4—treatment 4.

As expected, in most years, manual pruning incurred higher costs than mechanical interventions. Only in 2014 did interventions with the disc-saw pruning machine, comprising topping and hedging of both sides (T4), have costs similar to those performed with strictly manual pruning (T1).

In the case of mechanical pruning, costs depend on the number of pruning machine passages per tree line. One exception was the summer pruning performed in 2015 (T3 and T4), which had to be carried out at a lower speed, in order for the pruning machine to be able to cut the new flexible shoots in the upper part of the canopy.

It should be noted that the option of hedging only one side of the canopy resulted in the same costs as hedging both sides since the tractor has to make an empty path to restart the work always in the same direction.

When, in addition to topping and hedging, a manual complement (T2) was added (2014), pruning became more expensive than strictly manual pruning (T1).

In 2017, as a result of an intervention with more cuts by the tree, the work rate was lower than what had occurred in 2014. As a consequence, the cost weight of manual pruning gave rise to higher pruning costs of T1 and T2 treatments than those of T3 and T4 treatments.

Concerning total values, the cost of strictly mechanical pruning was lower than pruning with manual interventions. The option of pruning alternate faces every two years (T3) resulted in higher costs since more interventions were carried out.

Although only two pruning interventions were performed in T2 (2014 and 2017), the inclusion of a manual pruning complement resulted in a pruning cost similar to that obtained in T4 (only mechanical pruning).

### 3.3. Olive Yield

Olive yields were significantly ($p < 0.05$) influenced by the year (Table 1), revealing the common oscillation of this species. The significantly ($p < 0.05$) lower yield obtained in 2012 was due to the lack of irrigation water as a consequence of a severe drought. In 2014, a severe pruning intervention was carried out, with a considerable reduction in the canopy volume, contributing to a low level of olive production.

**Table 1.** Influence of the year on olive yield.

| Year | Yield (kg Tree$^{-1}$) |
| --- | --- |
| 2005 | 8.9 g |
| 2006 | 18.0 f |
| 2007 | 6.7 gh |
| 2008 | 25.8 cd |
| 2009 | 26.6 c |
| 2010 | 33.9 a |
| 2011 | 27.7 bc |
| 2012 | 2.0 i |
| 2013 | 20.4 ef |
| 2014 | 5.4 h |
| 2015 | 30.4 b |
| 2016 | 23.2 de |
| 2017 | 18.7 f |
| 2018 | 25.5 cd |
| Average | 19.5 |

Values followed by the same letter are not significantly different based on Tukey multiple-range test at the 5% level.

Table 2 shows olive yields by treatment for each year (between 2005 and 2009) and the average yield for this period. In 2005 and 2006, no significant differences ($p > 0.05$) were found in the olive yield per tree between treatments.

**Table 2.** Influence of the treatment on olive yield per tree between 2005 and 2009 (kg tree$^{-1}$).

| Treatment | 2005 | 2006 | 2007 | 2008 | 2009 | Average |
|-----------|------|------|------|------|------|---------|
| T1 | 8.7 ef | 1919 bc | 5.7 ef | 25.8 a | 25.3 a | 16.9 A |
| T2 | 8.8 ef | 18.5 c | 4.0 f | 25.6 a | 28.2 a | 17.0 A |
| T2 | 9.5 ef | 15.9 cd | 6.8 ef | 26.2 a | 24.8 ab | 16.6 A |
| T3 | 8.5 ef | 18.6 c | 10.4 de | 25.6 a | 28.1 a | 18.2 A |

Values with the same letter are not significantly different based on Tukey multiple-range test at the 5% level.

In 2007, significant differences ($p < 0.05$) were registered between treatments, with a significantly ($p < 0.05$) lower yield in treatment T2, compared with treatment T4. No significant differences ($p > 0.05$) were found between treatments in 2008, showing that the light pruning carried out with treatment T3 (topping and hedging on the east side of the canopy) did not influence the olive yield.

In 2009, no significant differences ($p > 0.05$) were found between treatments. On average, during the period 2005–2009, no significant differences ($p > 0.05$) were registered between treatments, which is associated with the low pruning intensity applied in the different interventions.

In the second period (2010–2013), significant differences ($p < 0.05$) were found in the effect of the treatments on yield. Table 3 shows olive yield by treatment in each year between 2010 and 2013, and the average yield for this period. In 2010, treatment T4 had a significantly ($p < 0.05$) lower yield than other treatments, which did not reveal significant differences ($p > 0.05$) between themselves. This can be explained by the reduction in the volume of the canopy imposed by pruning interventions (topping and hedging the two sides of the canopy).

**Table 3.** Influence of the treatment on olive yield per tree between 2010 and 2013 (kg tree$^{-1}$).

| Treatment | 2010 | 2011 | 2012 | 2013 | Average |
|-----------|------|------|------|------|---------|
| T1 | 34.3 ab | 28.1 cd | 1.4 g | 23.2 de | 21.8 AB |
| T2 | 38.1 a | 26.0 ce | 1.5 g | 21.8 e | 21.9 A |
| T2 | 38.1 a | 27.1 cd | 3.4 g | 10.9 f | 19.9 B |
| T3 | 25.2 ce | 29.4 bc | 1.7 g | 25.7 ce | 20.5 AB |

Values with the same letter are not significantly different based on Tukey multiple-range test at the 5% level.

In the following year (2011), no significant differences ($p > 0.05$) were found between treatments. In 2012, there were no significant differences ($p > 0.05$) between treatments in terms of olive yield, with the highest production resulting from treatment T3. There was a lack of irrigation water that year, so the smaller canopy size in treatment T3 (topping and hedging the west side of the canopy in 2012) may explain the result.

In 2013, there were significant differences ($p < 0.05$) between treatments in terms of olive yield; treatment T3 registered a significantly ($p < 0.05$) lower production than the other treatments, which did not differ from one another. The trees using treatment T3 had smaller canopy volumes than those using the other treatments because they were pruned in the previous year, while no pruning was performed in the other treatments since 2010.

On average, in this period (2010–2013), significant differences ($p < 0.05$) between treatments were found, with a significantly ($p < 0.05$) lower production in treatment T3 than in treatment T2.

It should be noted that in treatment T2, no pruning intervention was performed in this period: This shows that is possible to keep trees unpruned for some years, maintaining productive capacity, as already been seen in traditional olive orchards [4].

Table 4 shows olive yield by treatment in each year from 2014 to 2018 and the average yield of this period. In 2014, significant differences ($p < 0.05$) were registered in olive yield per tree between treatments. Treatment T3 revealed a significantly ($p < 0.05$) higher yield than the other treatments as a consequence of the smaller pruning intensity applied. In treatment T3, the significantly higher production may have resulted from the productive

branches issued in the previous year on the west side of the canopy, which was left uncut in 2014. In treatments T2 and T4, the lateral cuts on both sides of the canopy eliminated a considerable quantity of the productive branches issued in the previous year, diminishing productive potential in comparison with treatment T3. In the case of treatment T1, despite having been subjected to manual pruning, which has greater selectivity, the elimination of a considerable part of the canopy reduced the trees' productive potential.

**Table 4.** Influence of the treatment on olive yield per tree between 2014 and 2018 (kg tree$^{-1}$).

| Treatment | 2014 | 2015 | 2016 | 2017 | 2018 | Average |
|---|---|---|---|---|---|---|
| T1 | 34.3 hi | 28.1 a | 29.5 ab | 17.4 eg | 25.2 be | 22.0 A |
| T2 | 38.1 i | 26.0 ab | 26.6 ac | 16.2 eg | 24.9 be | 20.9 A |
| T2 | 38.1 gh | 27.1 bd | 18.5 dg | 19.9 cf | 26.5 ac | 20.3 A |
| T3 | 25.2 i | 29.4 ab | 18.0 dg | 21.3 cf | 25.4 bd | 19.9 A |

Values with the same letter are not significantly different based on Tukey multiple-range test at the 5% level.

In 2015, significant differences in olive yield were registered between treatments ($p < 0.05$). Treatment 3 obtained a significantly lower yield ($p < 0.05$) than the other treatments as a consequence of the higher yield obtained in the previous year. Given that the olive tree is characterised by an alternate bearing, a year with low production allows for more vegetative growth. Therefore, the greater leaf mass developed in one year will boost a higher yield the following year. This characteristic explains the considerable increase in production that occurred in treatments T1, T2, and T3 from 2014 to 2015. In the case of treatment T3, this increase was not so pronounced since, in 2014, the vegetative growth was conditioned by the existing tree production.

In 2016, significant differences ($p < 0.05$) were found in olive yield between treatments. Treatments T3 and T4 showed significantly ($p < 0.05$) lower yield than treatments T1 and T2. The fall in production of treatment T3, compared with treatments T1 and T2, is associated with a reduction in canopy volume due to topping in the summer of 2015 and cutting the west side of the canopy in the winter of 2016, which left these trees with lower leaf mass and, consequently, with lower fruiting potential. High production in a smaller tree canopy (e.g., T4 in 2015) tended to penalise the release of productive branches and, consequently, the production for 2016.

In 2017, production showed the opposite trend to that of 2016, although no significant differences ($p > 0.05$) between treatments were observed.

The absence of pruning interventions in 2018 contributed to no significant differences ($p > 0.05$) between treatments in olive yield in that year.

On average (Table 5), over the period 2005–2018, significant differences ($p < 0.05$) were found between treatments. Table 5 shows the average olive yield by treatment during this period. Treatment T3 registered a significantly ($p < 0.05$) lower yield than treatment T1 but showed no significant differences ($p > 0.05$) to the other treatments.

**Table 5.** Influence of the treatment on olive yield per tree on average, during a 14-year period.

| Treatment | Yield (kg tree$^{-1}$) |
|---|---|
| T1 | 20.2 A |
| T2 | 19.5 AB |
| T3 | 18.9 B |
| T4 | 19.4 AB |

Values with the same letter are not significantly different based on Tukey multiple-range test at the 5% level.

## 4. Discussion

These results showed that it is possible to maintain the productive level of high-density olive groves using only mechanical pruning interventions, as mentioned by [7], enabling pruning to be carried out quickly and with lower costs.

The option for more frequent mechanical pruning interventions (T3) tended to penalise olive yield and had higher costs. It can be concluded that performing a more intense mechanical pruning (T4) spaced in time is most desirable, as indicated by [7], for high-density olive groves. An irrigated olive grove in the phase of full production could maintain the productive level without performing pruning interventions, as shown by T2, during which trees were not pruned for 6 years. In [8], studying a traditional olive orchard, the authors have already verified that it is possible to spend 8 years without pruning, maintaining the average productive level of the grove. It should be noted, however, that this will lead to an increase in the size of the canopy, which could affect the harvesting process. Since the harvesting solution used in this study consisted of mechanical trunk shaking complemented manually with poles, larger trees required more harvesting time.

Although mentioned by some authors that a manual pruning intervention is required to free the canopy from excessive non-productive wood, this trial showed that manual complement to mechanised pruning, performed in the same year, did not provide any further advantages in olive yield, as verified by [8] in a traditional olive grove.

The advantage of performing manual pruning complement is related to the moment of its execution and the way in which it is carried out.

Ref. [9], studying a traditional olive orchard, verified that manual pruning complement used with mechanical pruning (T2), particularly in years following mechanical pruning, should be regarded as a potentially important technique, since it may contribute to higher yields after more than 10 years of submission to mechanical pruning.

In the case of this trial, two manual pruning interventions were executed over a period of 4 years. In both cases, the inner part of the canopy removed was higher than that in the traditional olive grove trial [9], in which just a few woody, non-productive branches were eliminated.

The opportunity for the use of manual pruning complement is a key question, and its periodicity should be addressed in further investigations.

However, the pruning strategy defined by [11] for traditional groves can be implemented in high-density olive groves, taking into account the higher vegetative growth of the latter. This strategy [9] consists of performing a horizontal cut of the canopy (mechanical pruning) at intervals of time that can stretch to 7 years. After the second intervention of mechanical pruning, a manual complement should be performed within the next two years. Then, after a new period of 5 to 6 years, a mechanical pruning intervention is performed, restarting the routine.

## 5. Conclusions

During this period of 14 years, on average, mechanical pruning tended to not influence olive yield negatively, except when executed with higher frequency.

Given this fact, the following conclusions can be drawn:

- The option of pruning with a disc-saw pruning machine on the side faces, in alternate years, is not a good solution since it does not obtain higher production of olives. Carrying out the cuts on the sides of the canopy more frequently while allowing the canopy volume to be controlled does not allow for the full-production potential of regrowth, which appears after the cuts on the side faces;
- Manual pruning complement is not useful since it does not lead to an increase in production but rather leads to an increase in pruning costs, as has already occurred in the traditional olive grove trial. Complementary manual pruning interventions only make sense for eliminating excess wood accumulated over relatively long periods of time and should be carried out sporadically. Two supplementary, annual pruning interventions over a period of four years are, therefore, not advisable.

**Author Contributions:** A.D. conceived and designed the experiment; A.D., J.F., A.P. and J.P. performed the evaluation; A.D. and J.P. analysed the data and wrote the paper. All authors have read and agreed to the published version of the manuscript.

**Funding:** This research is funded by National Funds through the Project PRODER entitled "Avaliação do desempenho da máquina de colheita contínua de azeitona (MCCA)", and Project PDR2020 entitled "Poda mecanizada e colheita em contínio de olivais de variedades portuguesas", both from the Ministry of Agriculture.

**Acknowledgments:** The authors acknowledge Torre das Figueiras Sociedade Agrícola Lda for the permission and availability of technical and human resources for the implementation of this study.

**Conflicts of Interest:** The authors declare that the research was conducted in the absence of any commercial or financial relationships that could be construed as a potential conflict of interest.

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
