# Peer review of "Effect of Mechanical Pruning on Olive Yield in a High-Density Olive Orchard: An Account of 14 Years"

_agronomy, doi:10.3390/agronomy12051105_

Round 1

Reviewer 1 Report

Overall view:

The most valuable thing about this experiment is its duration, 14 years, wich is an important issue to obtain conclusions in tree crops, and specially to evaluate pruning practices in olive trees, because these trees have reserves that can mask the differences between years.

But this long duration investigation means that the researchers, probably will have changed the experiment design along the years, for several and incontroled factors, like pluriannual funding ... and the differences between treatments are not easy to interpretate for  its low uniformity.

In any case, the overall conclussions obtained by the authors are usefull for the farmers.

Punctual comments:

Line 150: incomplete writing: '...was similar to that are normally...' perhaps you wish to say '...was similar to that normally...' ?

Line 152: Figure 2, in years 2014 and 2017 there are two columns/year, what means each col?

Line 154. '... exposure to cental...'  it should say 'exposure to central...'

Line 161, I think the following puntuations marks will help to understand the phrase: '... search work [1] and [2], a clarification...'

Line 165, it says '...to carried out...' it should say: '... to carry out..'

Line 166, the last words '....to be obtained' must be deleted.

Lines 172-181, can you give the information of tractor speed? It will improve the quality of the paper. Also it will be interesting to give a wider description of how the working rates were obtained: working speed, turning and other unproductive times, number of passes by row in each treatment,...

Lines 203-205, I suppose that the reported problem of the 'empty path' is debt to difficulties to change the position of the pruner arm when changing the direction of advance. This problem could be overcomed with a pruner with fast vertical side changing, or alterning along the years the hedging side, i.e., half of the trees with the East side hedged and the other half, the West side.  

Lines 230-240, although punctually some differences have been observed in some years, it is difficult to relate the differences to the pruning treatments, because the treatments are very similar in that period, T1 and T2 only differ in year 2005, and T3 and T4 in year 2008, so, with this conservative design, it is very difficult to evidence the differences between treatments.

line 384, it says 'A poda...' it may say: 'La poda...'

Results: there is only a table with the yields by year, but there is not any table with the yield by pruning treatment, just figures with significance letters, please include a table with the yield (kg tree-1) for each year and treatment.  

Discussion. It is difficult to obtain clear conclusions in an heterogenus experiment like this, with different treatments along the years. Perhaps, the most conveniente interpretation is to look to the overall yield after 14 years. Although no statistical differences have been found, it seams that manual pruning is the most productive treatment (figure 8). If we compare T3 and T4, the main difference is that in T3 there are 9 topping years front to 6 in treatment T4, perhaps this is the reason of the differences (no significatives at 5%) between these two treatments. Well, consider this just as some ideas in the case to follow with this experiments.

Good job!

Author Response

Response to Reviewer 1 Comments

Overall view:

The most valuable thing about this experiment is its duration, 14 years, wich is an important issue to obtain conclusions in tree crops, and specially to evaluate pruning practices in olive trees, because these trees have reserves that can mask the differences between years.

But this long duration investigation means that the researchers, probably will have changed the experiment design along the years, for several and incontroled factors, like pluriannual funding ... and the differences between treatments are not easy to interpretate for  its low uniformity.

In any case, the overall conclussions obtained by the authors are usefull for the farmers.

Response : I agree with the comment

Punctual comments:

Line 150: incomplete writing: '...was similar to that are normally...' perhaps you wish to say '...was similar to that normally...' ?

Response :  Corrected in the text

Line 152: Figure 2, in years 2014 and 2017 there are two columns/year, what means each col?

Response :  Corrected in the text

Line 154. '... exposure to cental...'  it should say 'exposure to central...'

Response :  Corrected in the text

Line 161, I think the following puntuations marks will help to understand the phrase: '... search work [1] and [2], a clarification...'

Response :  Corrected in the text

Line 165, it says '...to carried out...' it should say: '... to carry out..'

Response :  Corrected in the text

Line 166, the last words '....to be obtained' must be deleted.

Response :  Corrected in the text

Lines 172-181, can you give the information of tractor speed? It will improve the quality of the paper. Also it will be interesting to give a wider description of how the working rates were obtained: working speed, turning and other unproductive times, number of passes by row in each treatment,...

Response :  Corrected in the text

Lines 203-205, I suppose that the reported problem of the 'empty path' is debt to difficulties to change the position of the pruner arm when changing the direction of advance. This problem could be overcomed with a pruner with fast vertical side changing, or alterning along the years the hedging side, i.e., half of the trees with the East side hedged and the other half, the West side.  

Response :  I agree with the second part of your suggestion. In my opinion is preferably to make the hedging in all the orchard and then the topping. You don’t need to change  the position of the bar in each tree row. The execution of hedging before topping reduce the widht of the tree canopy which contribute to a better removal of the cut branches from the top of the canopy.

Lines 230-240, although punctually some differences have been observed in some years, it is difficult to relate the differences to the pruning treatments, because the treatments are very similar in that period, T1 and T2 only differ in year 2005, and T3 and T4 in year 2008, so, with this conservative design, it is very difficult to evidence the differences between treatments.

This trial was a learning process over time  with adjustment in the methodology according to the trees size, initially choosing to start with a conservative methodology.

Corrections in the text

line 384, it says 'A poda...' it may say: 'La poda...'

Response :  Corrected in the text

Results: there is only a table with the yields by year, but there is not any table with the yield by pruning treatment, just figures with significance letters, please include a table with the yield (kg tree-1) for each year and treatment.  

Response :  Corrected in the text

Discussion. It is difficult to obtain clear conclusions in an heterogenus experiment like this, with different treatments along the years. Perhaps, the most conveniente interpretation is to look to the overall yield after 14 years. Although no statistical differences have been found, it seams that manual pruning is the most productive treatment (figure 8). If we compare T3 and T4, the main difference is that in T3 there are 9 topping years front to 6 in treatment T4, perhaps this is the reason of the differences (no significatives at 5%) between these two treatments. Well, consider this just as some ideas in the case to follow with this experiments.

Reviewer 2 Report

In the reviewed manuscript entitled 'Effect of mechanical pruning on olive yield in a high density olive orchard - an account of 14 years' I propose a title change to "Effect of mechanical pruning on olive yield in a high density olive orchard".

The research presented in the paper has economic justification due to the reluctance of society to perform simple manual work. Hence, there is really a need to introduce mechanization where so far man seemed to be irreplaceable.

The Abstract was difficult for me to understand when I first read it, as it contains too much detail about the different treatments for trees. Its simplification can be considered, but a careful reader should understand the sense of the research and the results obtained by the authors. Due to the high level of complexity of this experiment, it is difficult to understand the individual treatments and the results obtained, but the authors presented them quite clearly as for their complexity.

I have doubts about the adequacy of the methods used to assess the amount of work, resources and the obtained yield, but I also do not know how it should be assessed in order for the obtained results to be more reliable and more scientific.

However, I understand that the main aim of the study, whether manual pruning of trees could be omitted, was achieved.

The method of citing publications should definitely be corrected, i.e. the information on who is the first author of the research should be added.

Are you sure there are no more publications on this topic? Seven items are not enough.

Linguistic errors should also be corrected.

In my opinion, please take into account the comments contained in the review and in the attached file. 

Author Response

Response to Reviewer 2 Comments

In the reviewed manuscript entitled 'Effect of mechanical pruning on olive yield in a high density olive orchard - an account of 14 years' I propose a title change to "Effect of mechanical pruning on olive yield in a high density olive orchard".

Response : In my opinion “an account of 14 years” is necessary to emphasise the long period of the trial

The research presented in the paper has economic justification due to the reluctance of society to perform simple manual work. Hence, there is really a need to introduce mechanization where so far man seemed to be irreplaceable.

Response: I agree

The Abstract was difficult for me to understand when I first read it, as it contains too much detail about the different treatments for trees. Its simplification can be considered, but a careful reader should understand the sense of the research and the results obtained by the authors. Due to the high level of complexity of this experiment, it is difficult to understand the individual treatments and the results obtained, but the authors presented them quite clearly as for their complexity.

Response: Since this is a long-term trial with many interventions over the years, I believe this is the best option to describe the work. The scheme presented in material and methods helps to realize the work done.

I have doubts about the adequacy of the methods used to assess the amount of work, resources and the obtained yield, but I also do not know how it should be assessed in order for the obtained results to be more reliable and more scientific.

Response: Some corrections were made

However, I understand that the main aim of the study, whether manual pruning of trees could be omitted, was achieved.

Response: I don’t understand the reason to omit the manual pruning

The method of citing publications should definitely be corrected, i.e. the information on who is the first author of the research should be added.

Response: Below you could see the rules of Agronomy

Are you sure there are no more publications on this topic? Seven items are not enough.

Response: There are a few more publications that shows the results of trials with a shorter number of years.

Linguistic errors should also be corrected.

In my opinion, please take into account the comments contained in the review and in the attached file. 

Response: corrected

Response to the comments:

Comment 1: I am not sure if this part of the title is needed.

Response: “an account of 14 years” is necessary to emphasise the long period of the trial

Comment 2: 1, 3 and 4 are the same Institute, so should be in one point, number 1. Please check where to add emails of all authors. Moreover, something is wrong with the font  style and size.

Response: Font style of the number 2 is different. Corrected in the text.

I think the number 3 and 4 appear because it was imported automatically when i put the name of the authors in the list

Comment 3: to

Response: corrected in the text

Comment 4: Which?

Response: The authors mentioned with number 2 in the references

Comment 5: surname should be added

Response: I think the rules of the Agronomy didn't require the name

Comment 6: second? What was the first?

Response: The first was the 8 years mentioned in the paragraph above

Comment 7: Who?

Response: Pastor and  Humanes. The same situation about the  editing rules

Comment 8: not in English. It has no sense.

Response: Corrected in the texto

Comment 9: Is this a citation? It should be rewrite.

Response: corrected in the text

Comment 10: east?

Response: West side. I understood your doubt, but the scheme was linked to the disposal of tree lines in the field.

Comment 11: west?

Response: east side. I understood your doubt, but the scheme was linked to the disposal of tree lines in the field.

Comment 12: ?

Response: age of the orchard

Comment 13: I don't understand this sentence.

Response: Pruners didn't perform manual pruning complement usually, and consequently they tend to make manual pruning. In manual pruning complement the objective is to make only 3 or 4 cuts inside the canopy. In manual pruning you remove more branches from the cnopy and consequently it is necessary to spend more time. As a consequence it is necessary to control the execution of manual pruning complement in order to obtain a real complement.

Comment 14: x man

Response: Corrected in the text.

In the legend is correct. Mechanical pruning is made by a man driving a tractor and consequently the work rate is always presented in trees/hour x man

Comment 15: longer

Response: I think it "require more time"

Comment 16: sound strange

Response: This is to put in evidence the results of treatment 3. Higher frequency. Corrected in the text

Reviewer 3 Report

The introduction should present the general state-of-art, highlighting the novelty and the importance of the present work. Are there similar studies for other fruit-trees orchards? not necessarily only for olive orchards... If yes, should be mentioned.

A total number of only 7 references seems very poorly. More references are needed and must be from other authors. I understand that authors worked many years on the presented research, but 6 self-citations from 7 references are really too much.

Author Response

Response to Reviewer 3 Comments

The introduction should present the general state-of-art, highlighting the novelty and the importance of the present work. Are there similar studies for other fruit-trees orchards? not necessarily only for olive orchards... If yes, should be mentioned.

A total number of only 7 references seems very poorly. More references are needed and must be from other authors. I understand that authors worked many years on the presented research, but 6 self-citations from 7 references are really too much.

Response : Some improvements were made, including more references to other trials made about the evaluation of mechanical pruning in olive orchards.
